# Enhancement of RGB-D Image Alignment Using Fiducial Markers

**DOI:** 10.3390/s20051497

**Published:** 2020-03-09

**Authors:** Tiago Madeira, Miguel Oliveira, Paulo Dias

**Affiliations:** 1Institute of Electronics and Informatics Engineering of Aveiro, University of Aveiro, 3810-193 Aveiro, Portugal; mriem@ua.pt (M.O.); paulo.dias@ua.pt (P.D.); 2Department of Mechanical Engineering, University of Aveiro, 3810-193 Aveiro, Portugal; 3Department of Electronics, Telecommunications and Informatics, University of Aveiro, 3810-193 Aveiro, Portugal

**Keywords:** computer vision, geometric optimization, camera calibration, 3D reconstruction, texture, inpainting, fiducial markers, point clouds, projection of 3D points

## Abstract

Three-dimensional (3D) reconstruction methods generate a 3D textured model from the combination of data from several captures. As such, the geometrical transformations between these captures are required. The process of computing or refining these transformations is referred to as alignment. It is often a difficult problem to handle, in particular due to a lack of accuracy in the matching of features. We propose an optimization framework that takes advantage of fiducial markers placed in the scene. Since these markers are robustly detected, the problem of incorrect matching of features is overcome. The proposed procedure is capable of enhancing the 3D models created using consumer level RGB-D hand-held cameras, reducing visual artefacts caused by misalignments. One problem inherent to this solution is that the scene is polluted by the markers. Therefore, a tool was developed to allow their removal from the texture of the scene. Results show that our optimization framework is able to significantly reduce alignment errors between captures, which results in visually appealing reconstructions. Furthermore, the markers used to enhance the alignment are seamlessly removed from the final model texture.

## 1. Introduction

Three-dimensional (3D) reconstruction is the creation of 3D models from the captured shape and appearance of real objects. It is a field that has its roots in several areas within computer vision and graphics, and has gained high importance in others, such as architecture [1], robotics [2], autonomous driving [3], medicine [4], agriculture [5], and archaeology [6]. Most of the current model acquisition technologies are based on LiDAR (Light Detection And Ranging) [7,8], RGB-D cameras [9,10], and image-based approaches, such as photogrammetry [11,12]. Despite the improvements that have been achieved, methods that rely on professional instruments and operation result in high costs, both capital and logistical [13].

The introduction of low-cost RGB-D cameras created an opportunity for 3D reconstruction of scenes to be performed at consumer-level. The increasing popularity of these sensors promoted the research of their use to reconstruct 3D indoor scenes [14,15,16]. Their manoeuvrability, permitting hand-held operation and allowing the user to get closer to parts of the scene and capturing them from different angles, proves to be an advantage in indoor environments with high probability of object occlusion, when compared with fixed high resolution scanners, which usually require more space and fixed poses to operate. As such, even though they are generally less accurate than fixed scanners such as LiDAR, mobile or hand-held scanners are known to be more suitable to perform indoor scanning [17]. These devices are also compelling for applications such as indoor navigation, since the way they are operated inherently conveys information about the empty space, which often corresponds to the navigable space, as the device is carried through it, by a person or a robot for instance.

Indoor 3D models have great potential in object tracking and interaction [18], scene understanding [19], virtual environment rendering [20], indoor localisation [21] and route planning [22], amongst others. Given the rapid development of LBS (Location-Based Services) and indoor applications, fast acquisition and high-fidelity reconstruction of complete indoor 3D scenes has become an important task [23]. Currently there is extensive research conducted into live 3D reconstruction techniques, where the final model is built and visualized during the data acquisition process. However, post-processing techniques show significant potential and need for future research, particularly in 3D reconstruction using RGB-D cameras [24]. As such, it would be interesting to complement the existing live 3D reconstruction techniques that use low-cost RGB-D hand-held cameras, considering their advantages and placement at consumer-level in the market, with some automatic post processing, to see whether their results can be improved, creating significant additional value.

The main objective of this work is to develop an optimization procedure capable of enhancing the 3D reconstructions created using a hand-held RGB-D camera, through the utilisation of fiducial markers placed in the environment.

## 2. Related Work

RGB-D cameras can obtain the depth of an object and the corresponding texture information, through the combination of a depth image and a standard RGB image. The depth images are commonly obtained using a ToF (Time-of-Flight) depth sensor measuring system [25] or Structured Light triangulation [26,27]. ToF works by measuring the round trip time of an artificial light signal, in this case IR (Infrared Radiation), to resolve the distance between the camera and the target object. Structured light works by projecting known patterns on to the scene and then measuring the way they are deformed, allowing for the calculation of depth and surface information. Modern RGB-D approaches are mostly based on the fundamental research by Curless and Levoy [28] who introduced the work of volumetric fusion, providing the foundation for the first real-time RGB-D reconstruction methods [29].

Registration is the process of alignment, in the same coordinate frames, of multiple captures, with different viewpoints, of the same scene or object. The result may be an extended version of a two-dimentional (2D) image, such as a panoramic photograph, or a 3D representation of the scene. Within the context of 3D reconstruction, registration translates to the problem of aligning the multiple point clouds that make up the 3D model of the scene, see Figure 1. When using RGB-D cameras, the dual nature of these sensors allows for the setup of the problem using two different approaches: registration can be computed by aligning several depth images or, conversely, the RGB images. Although either approach is feasible, in this work the RGB information was used for computing the alignment, since depth data had several disadvantages: lower resolution, relatively small precision, and sensitivity to lighting conditions.

Considering the example of an indoor reconstruction performed with a handheld device. This device would capture information as it travels through multiple positions in the environment, allowing for the collection of data from various places of the scene in different angles, places that may be occluded, or too far to be captured from the initial viewpoint of the acquisition. In order to obtain the completed 3D model, these multiple parts of the scene must be registered together. The alignment of point clouds is a problem directly tackled by ICP (Iterative Closest Point) [30]. However, this algorithm works by minimising the difference in a pair of point clouds. In order to align multiple ones, the algorithm must be applied many times. If this is done in a chain, for example, it creates a very real possibility of a drift, a cumulative error that grows as each pair is fit together.

In order to perform a registration, distinctive features of the captures, such as blobs, edges, contours, line intersections, or corners may be used to find correspondences between them. These features are typically represented in a feature vector. The selected features should be invariant to scale and rotation [31]. Examples are SIFT (Scale-Invariant Feature Transform) [32], SURF (Speeded Up Robust Features) [33], and HOG (Histogram of Oriented Gradients) [34]. The most common algorithms used in feature detection are Canny [35] and Sobel [36] for edge detection, Harris [37] and SUSAN [38] for edge and corner detection, Shi-Tomasi [39] and level curve curvature [40] for corner detection, FAST [41], Laplacian of Gaussian [42], and Difference of Gaussians [32] for corner and blob detection, MSER [43], PCBR [44], and Gray-level blobs [45] for blob detection.

Following the feature detection in a pair of captures, a set of key-points per capture is generated. Then, the matching of these key-points is performed, that is, identifying corresponding key-points between captures. There is a vast range of different approaches that tackle the matching problem, from brute-force matchers [46] and FLANN (Fast Library for Approximate Nearest Neighbours) [47] to pattern recognition [48], all of them very dependent on the type of scene and the available processing time. This step is still a challenge and there is always the possibility that a significant number of detections will be matched incorrectly.

As discussed previously, the feature matching process seldom works perfectly in real case scenarios. Matching features incorrectly means it is impossible to find a transformation that works for all matches found. One way to solve this problem is by recursively estimating a model that works for a subset of feature matches, until the size of that subset is considered large enough, RANSAC [49], while discarding the remaining matches, or even using the Iterative Closest Point (ICP) algorithm [30], that does not require individual feature matching in some implementations [50]. While there is still a matching procedure, it is merely based on distance. Each feature is paired with its closest neighbour of the reference capture and the transformation model is recursively estimated through a hill climbing algorithm, so the distance between neighbours approaches zero.

Some specialised software packages can automatically identify matching features in multiple images, such as Autodesk ReCap [51] or AliceVision Meshroom [52]. The distinctive features used are often corners or line segments. In the following step, instead of trying to align captures, the matching of the features is used to produce estimates of the camera positions and orientations (pose) and the 3D coordinates of these said features, producing a point cloud. It is important to refer that the better these estimates are, the better alignment of the captures we should expect to see, and as such the registration becomes an optimization problem.

In the context of this work, Visual Simultaneous Localisation and Mapping (Visual SLAM) [53] may be used. It utilises 3D vision to perform location and mapping, by solving an optimisation problem where the goal is to compute the configuration of camera poses and point positions that minimises the average reprojection error. The method of choice to solve this problem is called bundle adjustment, a nonlinear least squares algorithm which, given a suitable starting configuration, iteratively approaches the minimum error for the whole system.

Given a set of measured image feature locations and correspondences between them, the goal of bundle adjustment is to find 3D point positions and camera parameters that minimise the reprojection error. This optimization problem is usually formulated as a non-linear least squares problem, where the error is the squared $\ell^{2}$ norm, or Euclidean norm, of the difference between the observed feature key-points in the image and the projection of the corresponding 3D points to the image plane of the camera (reprojection error). The LM (Levenberg-Marquardt) algorithm [54] is the most popular algorithm for bundle adjustment.

In 2010, Agarwal et. al. [55] presented the design and implementation of a new inexact Newton type Bundle Adjustment algorithm. Considering the existence of a series of 3D points in the real world, these points are captured in images by different cameras, each camera being defined by its orientation and translation relative to a reference frame, its focal length and distortion parameters. After the desired acquisitions have been completed, the 3D points are projected into the images and the 2D coordinates are then compared to the ones obtained by feature detection in the images. The goal being to adjust the initial estimation of the camera parameters and the position of the points in order to minimise the reprojection errors, that is,
(1)$$\min_{{\hat{P}}^{i},{\hat{X}}_{j}}\sum_{ij}\ell^{2}{\left({\hat{P}}^{i}{\hat{X}}_{j},x_{j}^{i}\right)}^{2},$$
where $\ell^{2}$ is the Euclidean norm, $x_{j}^{i}$ are the coordinates of the *j*-th point as seen by the *i*-th camera, ${\hat{P}}^{i}$ is the projection matrix of the *i*-th camera, and ${\hat{X}}_{j}$ are the 3D points [56].

Another technique of which a key component is Bundle adjustment is Structure from Motion (SfM). SfM solves a problem analogous to visual SLAM, the main difference being that SLAM is usually meant to work in real-time on an ordered sequence of images, while SfM approaches often work on an unordered set of images as post-processing, many times done in the cloud. It can, using several images captured by one or multiple cameras, produce point cloud based 3D models, similar to those obtained by RGB-D cameras or LiDAR. This technique can be used to create models of objects with consumer-grade digital cameras and has been made possible by advances in computers, digital cameras, and UAV (Unmanned Aerial Vehicles). Together, these advances have made it feasible for a wide range of users to be able to generate 3D models, without extensive expertise or expensive equipment. SfM uses triangulation to calculate the relative 3D positions (X,Y,Z) of objects from pairs of images, often captured from a single moving camera, or with different cameras, in different locations and/or angles.

One of the main problems of registration, as discussed previously, is the unreliability of the detected features between captures. In the case of Bundle Adjustment, the minimisation of the reprojection may be computed using a wrongly matched feature in a pair of images, causing errors in the alignment of captures. Our approach uses fiducial markers to improve the feature detection and matching step. Markers are placed on the environment as visual features to be detected in post-processing. Aruco markers [57,58] were used (binary square fiducial markers) since existing software allows for robust and fast detection. Each marker has an identifier (id) (see Figure 2), making it very hard for false matching to occur: although under poor lighting conditions or in blurred images some aruco markers may go undetected, it is highly improbable that they will be identified with the wrong id.

The biggest downside to this approach is the pollution of the acquired scene’s texture with the fiducial markers. Therefore, along with the optimization procedure, an additional tool was developed, that allows the automatic removal of aruco markers from the texture of the scene, taking advantage of the obtained registration between acquisitions.

## 3. Optimization of Camera Pose Estimation Using Fiducial Markers

This section focuses on an optimization module developed to improve the registration of 3D reconstructions using fiducial markers. The problem was approached by creating a generic optimization Application Programming Interface (API) and applying it to the specific case of refining the pose of each camera with respect to a common reference frame in a 3D reconstruction procedure.

A camera is defined by the sensor’s internal properties and its positioning in space, relative to a reference frame. The former is represented by the intrinsic camera matrix, which was always consistent since the same device was used for all captures. It was retrieved using a chessboard based camera calibration procedure and can be represented as
(2)$$K=\left[\begin{array}{ccc} f_{x} & 0 & c_{x}\\ 0 & f_{y} & c_{y}\\ 0 & 0 & 1 \end{array}\right],$$
where $c_{x},c_{y}$ are the principal point coordinates, and $f_{x},f_{y}$ are the focal lengths expressed in pixel units. The latter is represented by an extrinsic camera matrix, such as
(3)$$T=\left[\begin{array}{cccc} r_{11} & r_{12} & r_{13} & t_{x}\\ r_{21} & r_{22} & r_{23} & t_{y}\\ r_{31} & r_{32} & r_{33} & t_{z} \end{array}\right],$$
which may be decomposed into two components, a translation and a rotation. The translation can be represented as the vector $T=(t_{x},t_{y},t_{z})$, and the rotation can be represented as a 3×3 rotation matrix *R*.

The optimization is performed as a bundle adjustment, meaning the objective of the optimization is to refine the camera poses and the 3D point positions. As such, the set of parameters to be optimized Φ, is defined as
(4)$$\begin{array}{c} \Phi =[\;\overset{Camera\;poses}{\overset{︷}{x_{i=1},y_{i=1},z_{i=1},r1_{i=1},r2_{i=1},r3_{i=1},\ldots ,x_{i=I},y_{i=I},z_{i=I},r1_{i=I},r2_{i=I},r3_{i=I},}}\\ \overset{Marker\;translations}{\overset{︷}{x_{j=1},y_{j=1},z_{j=1},\ldots ,x_{j=J},y_{j=J},z_{j=J}}}\;]\;, \end{array}$$
where *i* refers to the *i*-th camera, of the set of *I* cameras, and *j* refers to the *j*-th aruco marker, of the set of *J* aruco markers. Notice that, in this vector, the rotation for one camera ($r_{1},r_{2},r_{3}$) is represented through the axis/angle parameterization, as opposed to the 3×3 rotation matrix format of the camera’s extrinsic matrix. Because a rotation matrix has 3×3=9 elements, but only 3 degrees of freedom, a different parameterization is necessary in order to intrinsically incorporate constraints on the rotations during the optimization.

Popular parameterization for rotations are Euler angles, quaternions, and axis/angle representation. However, not all representations are suitable for an optimization. Parameterization should not introduce more numerical sensitivity than the one inherent to the problem itself, as this decreases the chances of convergence in optimizations. When the parameterization formats follow this rule, they can be referred to as fair parameterization [59]. For example, Euler angles, which are probably the most used angle parameterization, are not suitable for optimizations [60], because they do not yield smooth movements, each rotation is non-unique and, most notably, they introduce singularities, known as Gimbal lock, where one degree of freedom is lost [60]. Because quaternions have 4 components which are norm-1 constrained, and this introduces some complexity in the algorithms, they are usually not used for optimizations [60], even though they are a fair parameterization. The axis/angle parameterization is the most widely used to represent a rotation in an optimization. It is a fair parameterization and has only three components, two values to define a unit vector indicating the direction of an axis of rotation, and one value to define an angle. In this way, any rotation can be represented as a rotation around this axis, by an angle θ. To convert between the axis/angle format and the 3×3 rotation matrix format (used in the data model to ease implementation) the Rodrigues’ rotation formula was used.

The datasets were obtained using the Google Tango platform, meaning they already contained information about the poses of the cameras, which was used as a first guess for the optimization procedure. Thus, in this sense, the proposed approach is a refinement of an initial proposal of camera pose and corresponding reconstructed digital 3D model. However, if this data is not available, our system is able to produce one by utilizing the detected markers. In this case, one of the cameras is considered as the World reference frame and the transformations from each camera to the World are calculated using chains of transformations between markers and cameras, taking advantage of common marker detections between cameras. We also needed a first guess for the positions of the 3D points, which correspond to the aruco marker centers. Their poses must be determined in the world reference frame. Having an initial guess of the camera poses (transformations from each camera to the world), for each aruco marker we need only a transformation from its reference frame to one of the cameras in which it can be detected. Thus, the initial positions for the 3D points are obtained by calculating the aggregate transformation from each aruco marker to the world reference frame as
(5)$$\begin{array}{c} {}^{A_{j}}T_{W}\;={\;}^{C_{i}}T_{W}\;\cdot {\;}^{A_{j}}T_{C_{i}} \end{array}\;,$$
where $A_{j}$ refers to the *j*-th aruco marker detected, $C_{i}$ refers to the *i*-th camera (the first in which the *j*-th aruco marker can be detected), and *W* refers to the world reference frame.

The cost function is based on the reprojection error, that is, the geometric error corresponding to the distance (in pixels) between a 3D projected point (with the evaluated pose) and a measured one (computed from the aruco detection), as in Bundle Adjustment. The relationship between a 3D point in the world and the pixels that correspond to its projection on an image plane can be expressed as
(6)$$s\left[\begin{array}{c} u\\ v\\ 1 \end{array}\right]=K\;T\left[\begin{array}{c} X\\ Y\\ Z\\ 1 \end{array}\right],$$
where X,Y,Z are the coordinates of a 3D point in the world; u,v are the coordinates of the projection point in pixels; *K* is the intrinsic camera matrix and *T* is the extrinsic camera matrix. *T* translates coordinates of a 3D point (X,Y,Z) to a coordinate system fixed with respect to the camera, meaning it corresponds to the transformation from the world to the camera reference frame.

The optimization is performed using a nonlinear least-squares regression, as indicated for Bundle Adjustment problems. The function *least squares* from the SciPy library [61] is used. It requires a cost function, a vector of parameters, bounds (infinite by default), and a sparse matrix (initializes to identity if not available) to carry out the optimization. At the end of the optimization, the function returns a vector with the optimized parameters for the pose of the cameras and the position of the 3D points, which minimises the reprojection errors. Because of the wrapper implemented, these values are automatically converted and copied from this vector to their place in the data models.

The visualisation function for this optimization consists of showing all the images, each with the aruco markers centers detected, the initial projections connected to the target by a blue line, and the projections at the current step of the optimization, as seen in Figure 3 on the left. In this case, it is the end of the optimization, and so the final position is overlapping with the detected one. This is complemented by a 3D representation of the pose of the cameras and position of the 3D points in the scene, showcased in Figure 3 on the right.

The estimated transforms are applied to the point clouds to ensure that the optimized datasets are in the calibrated reference frame. The applied transformation is
(7)$$\begin{array}{c} {}^{\mathrm{oldW}}T_{\mathrm{newW}}\;={\;}^{C_{i}}T_{\mathrm{newW}}\;\cdot {\;}^{\mathrm{oldW}}T_{C_{i}}\; \end{array}\;,$$
where oldW refers to the world reference frame before optimization, newW refers to the world reference frame after optimization, and $C_{i}$ refers to the *i*-th camera.

In Figure 4, we can see a merged point cloud coloured with the original images. The enhancement of the registration caused by the optimization improves the texture significantly, as showcased by Figure 5 in further detail. Figure 6 presents the results of the process in several views, showing clearly a better alignment of the different point clouds after the optimisation.

## 4. Texture Refinement and Marker Removal

This section focuses on a module developed to automatically remove fiducial markers from a scene. In Section 3, a method to improve the registration of a 3D reconstruction using fiducial markers was presented. The module showcased in the present section was created as an attempt to avoid the pollution of the acquired scene’s texture with these markers. There are other scenarios, besides 3D reconstruction, where having fiducial markers in the environment is advantageous, such as Augmented Reality (AR) applications, and being able to successfully remove them from the texture may improve the final visualizations significantly.

This module is agnostic to the way the dataset is stored and loaded to memory. By using a different dataset loader, one may utilise this inpainting tool in a different context. The tool was programmed to detect and remove aruco markers, since this was the kind of fiducial marker used in Section 3. However, it could be used for different markers by adapting the detection and pose estimation functions. All the masks showcased in this section were blended with images to better showcase where they fit and what regions are the target.

### 4.1. Navier-Stokes-Based Inpainting

The Navier-Stokes-based Inpainting algorithm available in the OpenCV library was used in this work. Its function is to restore a region in an image using the region neighbourhood. To operate, this algorithm requires a mask representing the region to replace. In a first attempt, the masks of the fiducial markers were created using the corners provided by the aruco detector. This method was not valid since these masks did not account for the margin around the markers (see Figure 7 left), resulting in blank cards scattered through the scene. To solve this issue, the first approach taken was to dilate the masks. However, this 2D operation in the image plane did not solve the problem for markers that are close or with large angles relative to the camera (see Figure 7 center). The solution found was to create a mask as a 3D object, with the same dimensions as the markers (including thickness), and projecting the 3D masks, corresponding to each detected marker, to the 2D images. This ensures the whole intended area is masked at any distance and angle (see Figure 7 right).

Having an accurate representation of the region covered by the marker, the challenge was to define the colours that will be used to restore that region. The results of the Navier-Stokes-based Inpainting from OpenCV were underwhelming, as shown in Figure 8. The examples found online for the testing of this algorithm are usually about removing lines from the images or restoring damage from folding photographs. Inpainting in this manner is usually done in small areas, while the size of a fiducial marker in the scene is substantial.

#### 4.1.1. Inpaiting of Homogeneous Regions through Image Blurring

Given the limitation of the first implementation of the inpainting, and taking into account the placement of the markers over homogeneous texture regions, the blurring of the area where the inpainting is applied was performed with high aperture linear size (median blur with kernel size of 201), as seen in Figure 9.

The texture obtained after blurring (Figure 9) presents some improvements, when compared to the inpainting restoration (Figure 8). However, the limits of the inpainted areas are still sharp and visible. To reduce this effect, blurring is applied once again, this time in a bigger area, defined by enlarging the 3D masks of the markers and projecting them in the obtained images (Figure 9), now using a smaller aperture linear size for the blur (median blur with with kernel size of 51), to avoid mixing colour of the surrounding objects, see (Figure 10).

At the end, a bilateral filtering is applied to preserve the edges in the image, avoiding the mixing of colours, while smoothing over the surfaces, making the grainy pattern of the walls less noticeable for example, see Figure 11.

The values for the blurring factor, along with the size of the marker objects and patch size around them, can be modified through command line arguments or assigned easily in the code. These values were empirically set because they depend on various distinct factors, the characteristics of the scene, and the sensors used.

#### 4.1.2. Inpainting Non-Detected Markers: Cross-Inpainting

It is frequent to miss fiducial marker detections in images, when the marker is not fully visible (for example in the border of the image), or when blurring occurs due to motion, for instance. Since the transformations from the aruco markers to cameras and from the cameras to the world are known (see Figure 12), it is possible to evaluate the presence of a marker in a given image, even if the marker is not detected in that particular image.

The solution implemented, named “Cross-Inpainting”, takes advantage of transformations from the aruco markers to the world. Every time a new marker is detected in one of the images of the dataset, the transformation of the marker to the world is known. This transformation is computed using the transformation of the marker to the camera where it was detected, the *i*-th camera, and the transformation from that camera to the world:(8)$$\begin{array}{c} {}^{A_{j}}T_{W}\;={\;}^{C_{i}}T_{W}\;\cdot {\;}^{A_{j}}T_{C_{i}} \end{array}\;,$$
where $A_{j}$ refers to the *j*-th aruco marker detected, $C_{i}$ refers to the *i*-th camera, and W refers to the world reference frame. Given an image captured by a camera *k*, when this image is being inpainted, it is possible to access the information for all markers and check if a marker should be present in the image even if not detected. This is done by applying the transformation from the aruco marker reference frame to the world, as in Equation (Equation 8), followed by the transformation from the world to camera *k*:(9)$$\begin{array}{c} {}^{A_{j}}T_{C_{k}}\;={\;}^{W}T_{C_{k}}\;\cdot {\;}^{A_{j}}T_{W} \end{array}\;,$$
where $A_{j}$ refers to the *j*-th aruco marker, $C_{k}$ refers to the *k*-th camera, and W refers to the world reference frame. A 3D object representing the marker is transformed using Equation (Equation 9) and then projected to the image captured by camera *k*. This idea was applied to non-detected aruco markers before optimization (Figure 13 left) and after optimization (Figure 13 right), showing the viability of this approach after optimization to improve the inpainting of partially visible or non detected aruco markers, see Figure 14.

#### 4.1.3. 3D Point Cloud with RGB Visualisation

To better visualise and evaluate the results of the registration and inpainting operations, we developed operations to allow the visualization of fused RGB coloured 3D point clouds. For each point cloud, the coordinates of the 3D points are projected into the corresponding colour images to extract RGB pixel values. Finally, a new .ply file is created with the XYZ-RGB information for the fused point cloud that can be visualized in any 3D viewer (Meshlab, PCL viewer, PPTK library, etc.). An example of the registered visualization of 10 coloured point clouds with inpainting, before and after optimization, is presented in Figure 15.

For comparison purposes, in Figure 16 a merged point cloud, coloured with images restaured using only the detected markers can be observed. At first glance, the texture applied before optimization seems better, this effect is caused by the fact that the points closest to the viewer happen to correspond to images where more markers were detected. After the optimization, the merged point clouds are better aligned and the undetected markers, which in Figure 15 (right) were removed using cross-inpainting, here show through in Figure 16 (right).

## 5. Results

This section contains the evaluation methodology and the description of its process, going over the collection of datasets and showcasing the results obtained in various ways.

### 5.1. Evaluation Methodology

Visual comparison of the point clouds may sometimes prove to be somewhat subjective and inconclusive. Particularly in datasets with a very large number of point clouds. The reporting of results in a document also means that the changes must be observed in a side by side manner, which makes them harder to perceive than in a 3D visualization software such as Meshlab, for instance, where it is possible to instantly switch between point clouds placed in the same exact place. In order to provide some quantitative results, we performed experiments in a meeting room of the Department of Mechanical Engineering (DEM) at the University of Aveiro, for which a laser scan was acquired with a FARO Focus Laser Scanner [62]. Given the high precision of the FARO laser sensor, a point cloud of the scan was considered as the ground truth to evaluate the proposed process of registration based on fiducial markers, and compare it to a Google Tango reconstruction [63], as well as a reconstruction with registration refinement using ICP [30]. As the meeting room was larger than the previously used datasets, it was also an opportunity to create a more difficult challenge for the optimizer.

### 5.2. Datasets

The datasets were collected using the ZenFone AR, a Google Tango enabled Android phone, equipped with an RGB-D camera. The example dataset, used in previous sections, was collected in office 22.2.18 of DEM: a relatively small dataset with a large density of fiducial markers. Two additional, more complex datasets were captured: the meeting room mentioned in Section 5.1, and a lobby, just outside the meeting room. Unfortunately, due to logistic problems, the FARO laser scan and the RGB-D datasets were not acquired in the same day, resulting in some differences in the scene (namely some furniture had been moved within the room, such as couches and chairs). Despite this limitation, it was possible to validate our results using an area of the model with little change and focusing on walls and ceilings.

### 5.3. Collecting Ground Truth

The laser scan of the meeting room was acquired with a FARO Focus Laser Scanner [62]. A single point cloud was obtained from the merging of several scans, in Cloud Compare. However, given its very high density (50698121 vertices), unnecessary for the purpose of this study, a downsampled and clipped version was generated using Cloud Compare. The downsampling of the FARO laser acquisition result (3722614 vertices) can be observed in Figure 17, and the final result (1777550 vertices) in Figure 18.

### 5.4. Evaluation Procedure

The key metric used for the evaluation procedure was the Hausdorff distance. It measures how far one subset of a metric space is from the other, and is often applied to the measurement of distance between point clouds. It can be thought of as the maximum distance of a set to the nearest point in the other set [64]. The Hausdorff distance from set A to set B may be defined as
(10)$$h\left(A,B\right)=\underset{a\in A}{\sup}\underset{b\in B}{\inf}d\left(a,b\right)\;,$$
where sup represents the supremum, inf represents the infimum, and d(a,b) is the Euclidean distance between a and b.

The tools available in Meshlab, allow us to compute, not only an absolute distance, but also the distance at each point of the cloud. This distance can then be mapped visually through a colour. ICP was used in Cloud Compare to align processed point clouds with the FARO laser ground truth point cloud. Then, Meshlab’s Haudorff distance was used to compute the distance between the given point cloud and ground truth, providing a coloured visualization of the distance. A red-yellow-green-blue colourmap was used, where red maps a small distance and blue a large distance. This visualization allows for the understanding of which areas have significant error compared to the ground truth. It also enables the identification of areas where the scene had physically changed between the capture of the ground truth and our datasets (namely furniture movements).

We used Google Tango’s reconstruction as baseline and compare it with a cumulative Iterative Closest Point (ICP) approach, and our proposal. The cumulative ICP approach was applied in a pairwise fashion from cloud i to i+1, these captures were then merged and became cloud i for the next ICP calculation. The starting point of the reconstruction for the ICP procedure was the same as the one used for our approach.

### 5.5. MeetingRoom Dataset

In this subsection, the results obtained using the *MeetingRoom* dataset are presented. This dataset takes advantage of the fact that there is the possibility of analysing it in comparison with the FARO laser scanned point cloud, see Section 5.3. The results are compared with Google Tango and ICP, first using the texture of the original images in Figure 19, Figure 20 and Figure 21, then using the Hausdorff distance based color mapping in Figure 22 and Figure 23, and finally, the fiducial marker removal results are showcased in Figure 24.

The *MeetingRoom* dataset includes 136 RGB-D images and the alignment optimization was performed in 3m50s on an Ubuntu 18.04 laptop equipped with an Intel Core i7 7500U CPU and 8GB of RAM, see Table 1.

The Hausdorff distance was calculated from each of the methods’ produced reconstruction to the FARO laser generated point cloud. The local values were used to colourmap the point clouds, see Figure 22 and Figure 23. The mean and Root Mean Square (RMS) values, see Table 2, indicate an improvement of approximately 27% of our approach over Google Tango. Regarding the comparison of our approach with ICP, it shows an improvement of approximately 13% in RMS and 5% in mean. The maximum distance considered for the measurement was empirically set to 15 cm. These values should be considered with care because they are computed globally for a very large number of points, many of which were not moved significantly during the optimization since multiple areas are already close to the ground truth before the optimization. As a result, areas that improved significantly may not have a very high impact on the value of the mean and RMS. The RMS is usually considered a more meaningful metric in the context of 3D reconstruction, since it gives a relatively high weight to large errors, which appear in the form of visual artifacts (see Figure 23).

The main identified limitations of our approach are: the overhead of time and logistics associated with the marker placement in the scene, when compared with methods that do not require the preparation of the environment previous to the capturing process, which is very dependent on the user, nature of the scene, and the density of markers desired; the need for at least one marker to be detected in a capture, ideally more, to allow for its registration refinement; and the pollution of the scene’s texture. Some of the observed texture problems in the ICP results may be attributed to the fact that a number of captures contain points mostly belonging to the same plane, and in a pair of those captures, an ICP procedure may slide them (within the plane), while maintaining a low distance between the captures. Our approach does not suffer from this problem.

Table 1 presents the 3D alignment processing times for the MeetingRoom dataset using ICP and our approach. Google Tango processing times are not included, since it performs bundle adjustment optimization during the capture process and thus it is not directly comparable with our method that only executes the optimization after all the data has been captured. It is noteworthy that Google Tango also requires additional time after data capture to produce the final reconstructed model. Unfortunately, it is difficult to evaluate precisely tango processing time since the system is not open source and the time we could evaluate would also include other processes besides the optimization, such as mesh creation and processing of stored data changes.

### 5.6. Lobby Dataset

In this subsection, the results obtained using the *Lobby* dataset are showcased. This dataset is analysed through visual comparison with the Google Tango reconstruction, starting with the texture of the original images in Figure 25 and Figure 26, then showing some detail using a colourmap, see Figure 27, and finally showcasing the fiducial marker removal results in Figure 28.

The *Lobby* dataset includes 84 RGB-D images and the alignment optimization was performed in 1m05s on an Ubuntu 18.04 laptop equipped with an Intel Core i7 7500U CPU and 8GB of RAM.

## 6. Conclusions

It is possible to observe a significant improvement in the registration of the 3D points clouds using our marker based optimization approach. Nevertheless, there are some limitations derived from the fact that this optimization only targets the registration problem, while having no influence on the geometry of each point cloud. In other words, the alignment of the point clouds may be improved, but if the point clouds are themselves distorted, which might occur with RGB-D data, the geometry of the scene may not improve significantly from this optimization.

The optimization implemented attempts to improve the results obtained from a 3D reconstruction using RGB-D cameras. If the dataset obtained from the reconstruction has poor alignment of the point clouds, even if only on some areas, then we should expect the optimization to improve the scene’s geometry significantly. However, if a particular dataset happens to have very good alignment, the improvement will probably prove to be minor.

The API developed provides useful abstractions and an environment for future implementation of optimizations. It facilitates building upon and future work, allowing for the writing of more intuitive code. It also provides structure for a systematic approach to this kind of problem. We have already made use of these tools in other projects that require an optimization to be performed, such as the calibration of a set of sensor in an autonomous vehicle [65] and colour consistency correction in 3D reconstructions.

The removal of the fiducial markers from the texture of the scene was achieved, though it is important to note that the tool developed was meant for restoring homogeneous texture areas only, without a recognisable or sharp pattern. This tool allows for the creation of point clouds with texture free of markers, by utilising RGB-D information. The cross-inpainting technique was successful at removing non-detected aruco markers from the images, if there is good registration of the point clouds. For this reason, the technique also serves as a way to confirm that the optimization is working as intended, since its results, which depend on the good alignment of the point clouds, improve after the optimization is performed.

Results presented include several qualitative and quantitative assessments in three different datasets. Thus, we are convinced that the approach is robust and should in principle work for other datasets.

In future work, it would be interesting to generate meshes from the final point clouds, coloured with the texture free of fiducial markers. These meshes may be created through the utilization of ROS-based CHISEL [63], for instance.

There are some questions about the ideal density of fiducial markers in the scene, that is, how many markers should be spread, on average, to be seen in each image, to obtain the best results from the optimization. In the future, this could be analysed, producing guidelines regarding marker density and placement to optimize preparation time and alignment results.

Regarding the removal of fiducial markers, it would be interesting to investigate techniques that would allow the restoration of texture containing distinctive patterns, to enable its use in a wider range of environments and applications.

## Figures and Tables

**Figure 1 sensors-20-01497-f001:**
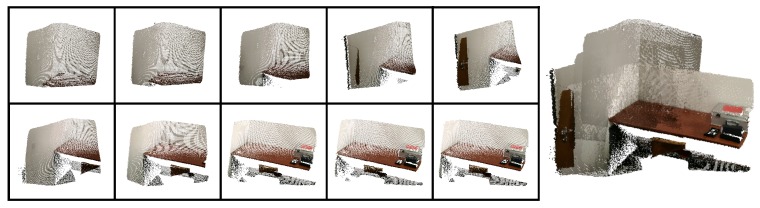
Example of registration of several point 3D point clouds to create a complete 3D model.

**Figure 2 sensors-20-01497-f002:**
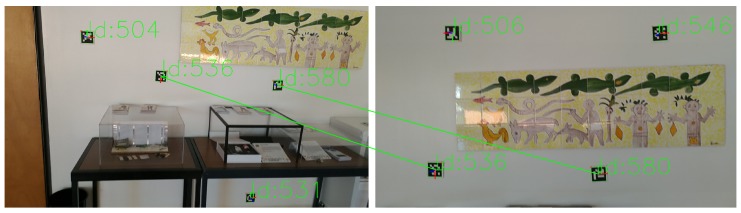
Detection and matching of aruco markers in a pair of captures.

**Figure 3 sensors-20-01497-f003:**
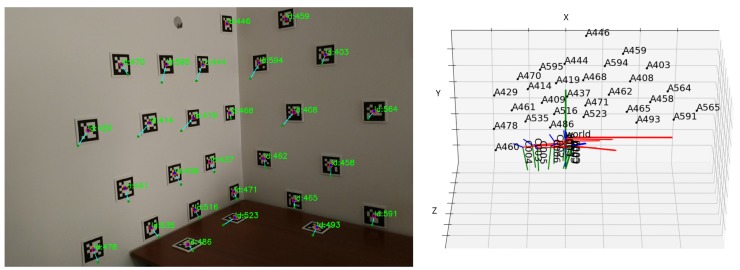
Visualisation function: (left) Aruco marker detected position in red (target of optimization), initial projection in green, current projection in dark blue, drawn over one image; (right) 3D representation of the position of the cameras and aruco markers in the scene.

**Figure 4 sensors-20-01497-f004:**
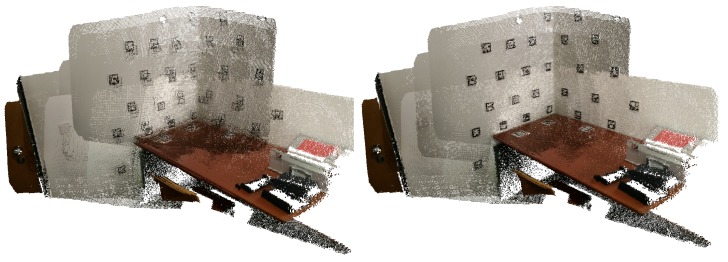
Merged point cloud coloured with texture obtained from images: (**left**) before optimization; (**right**) after optimization.

**Figure 5 sensors-20-01497-f005:**
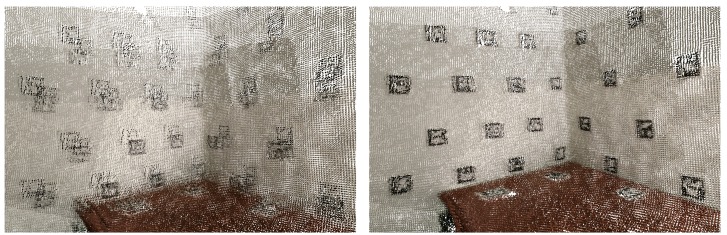
Merged point cloud coloured with texture obtained from images (detail): (**left**) before optimization; (**right**) after optimization.

**Figure 6 sensors-20-01497-f006:**
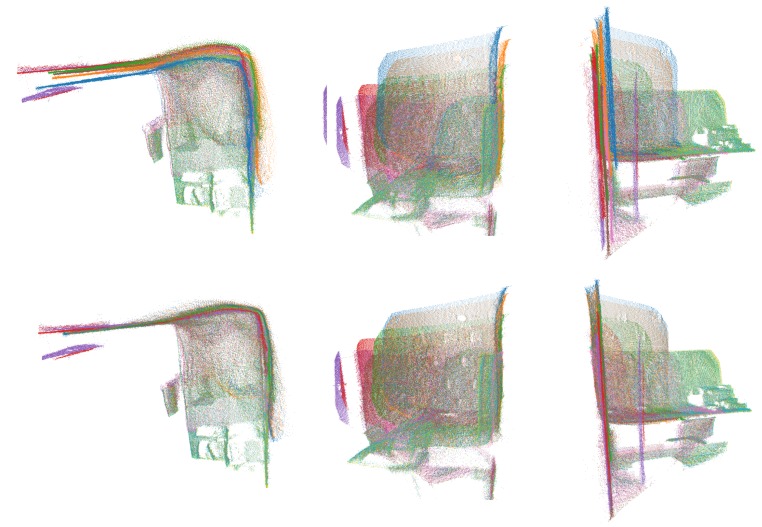
Several views of a reconstructed scene: (**top**) before optimization; (**bottom**) after optimization. Point clouds corresponding to each capture are colored differently for better visualization.

**Figure 7 sensors-20-01497-f007:**
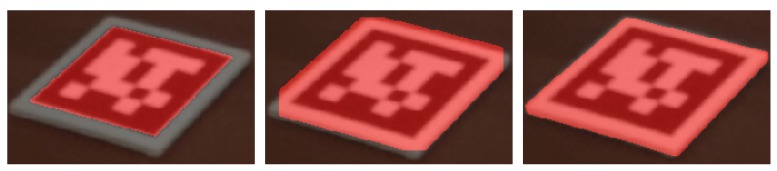
Methods for mask creation: (**left**) detection; (**center**) 2D dilation; (**right**) 3D projection.

**Figure 8 sensors-20-01497-f008:**
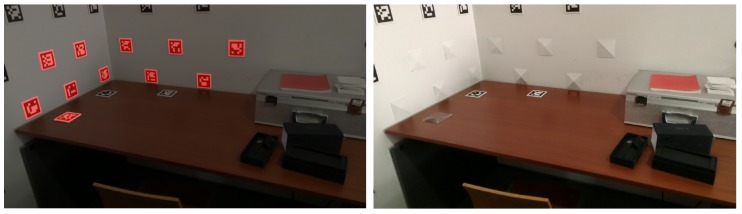
Example dataset image inpainting: (**left**) mask used for marker removal; (**right**) results of Navier-Stokes-based inpainting from OpenCV.

**Figure 9 sensors-20-01497-f009:**
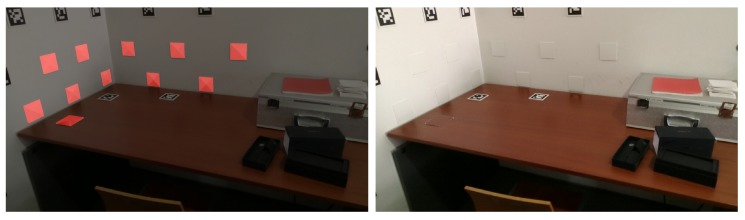
Example dataset image restoration attempt: (**left**) mask for the pixels to be replaced; (**right**) results with blurring.

**Figure 10 sensors-20-01497-f010:**
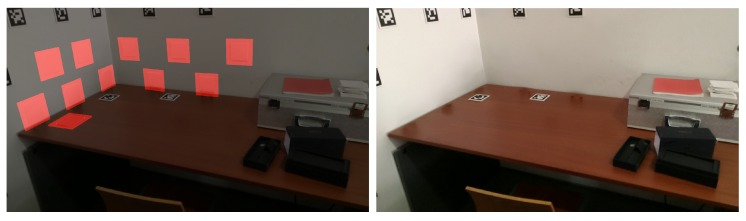
Example dataset image restoration improvement: (**left**) mask for the pixels to be replaced (based on enlarged 3D masks); (**right**) results with second blurring.

**Figure 11 sensors-20-01497-f011:**
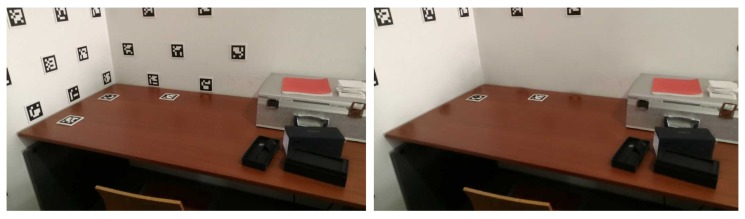
Example dataset image restoration: (**left**) original image; (**right**) restored image.

**Figure 12 sensors-20-01497-f012:**
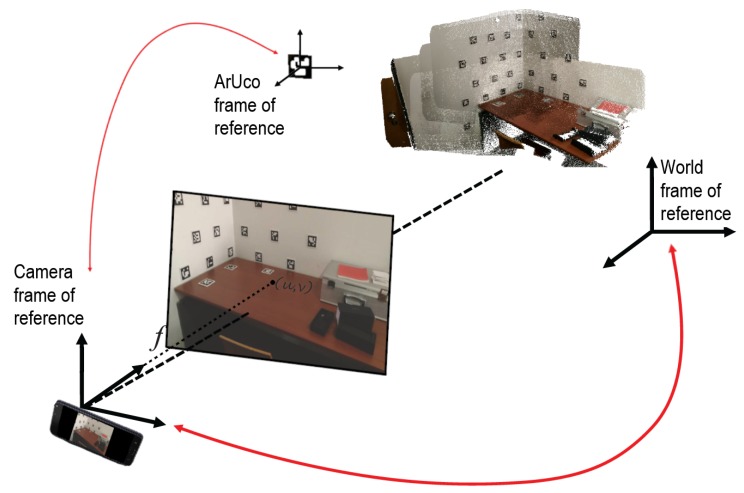
Conversion between different reference frames is used to perform cross-inpainting.

**Figure 13 sensors-20-01497-f013:**
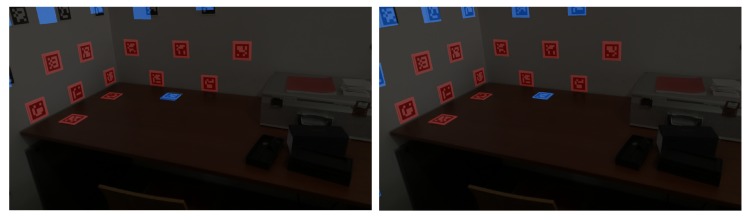
Masks obtained from aruco marker detection in red and masks obtained from projections from other cameras (cross-Inpainting) in blue: (**left**) before optimization; (**right**) after optimization.

**Figure 14 sensors-20-01497-f014:**
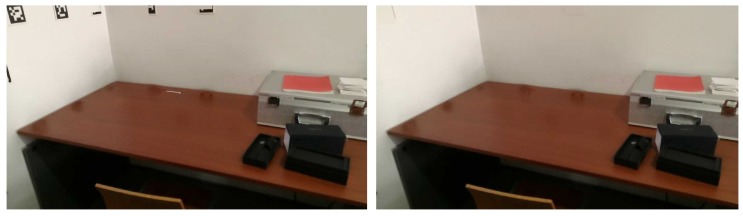
Restoration with cross-inpainting result: (**left**) before optimization; (**right**) after optimization.

**Figure 15 sensors-20-01497-f015:**
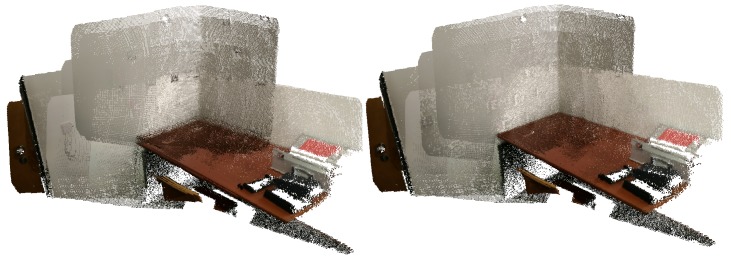
Point cloud coloured with texture obtained from restored images, using cross-inpainting: (**left**) before optimization; (**right**) after optimization.

**Figure 16 sensors-20-01497-f016:**
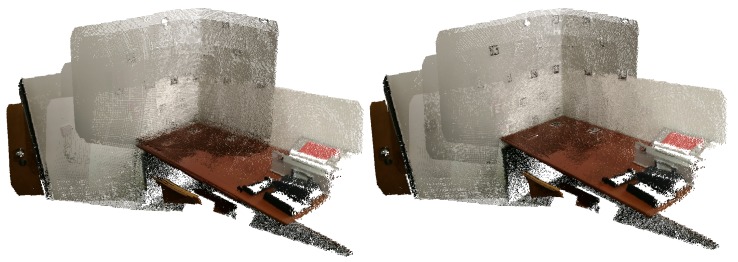
Point cloud coloured with texture obtained from restored images, no cross-inpainting: (**left**) before optimization; (**right**) after optimization.

**Figure 17 sensors-20-01497-f017:**
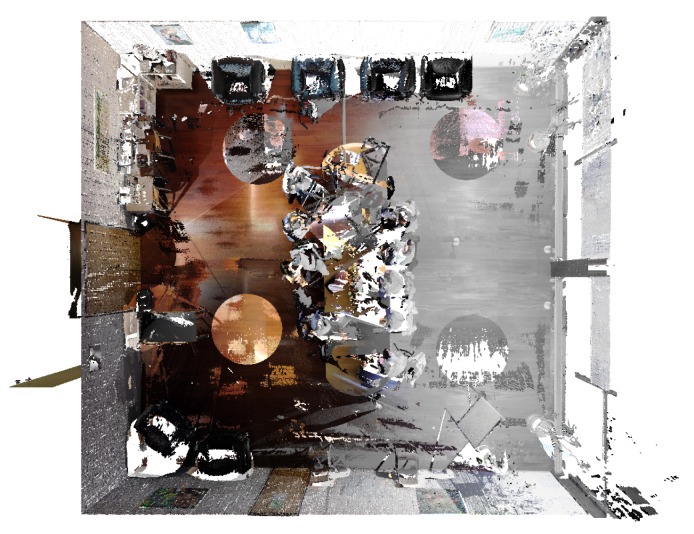
FARO laser scanned point cloud after downsampling.

**Figure 18 sensors-20-01497-f018:**
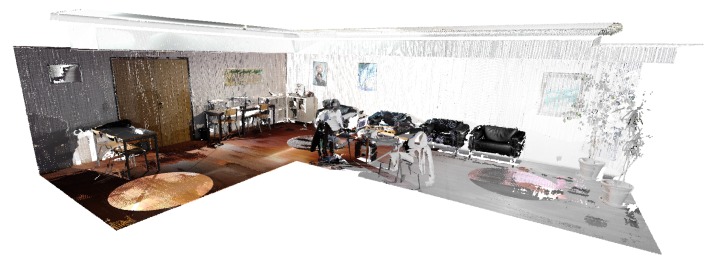
FARO laser scanned point cloud after downsampling and clipping.

**Figure 19 sensors-20-01497-f019:**
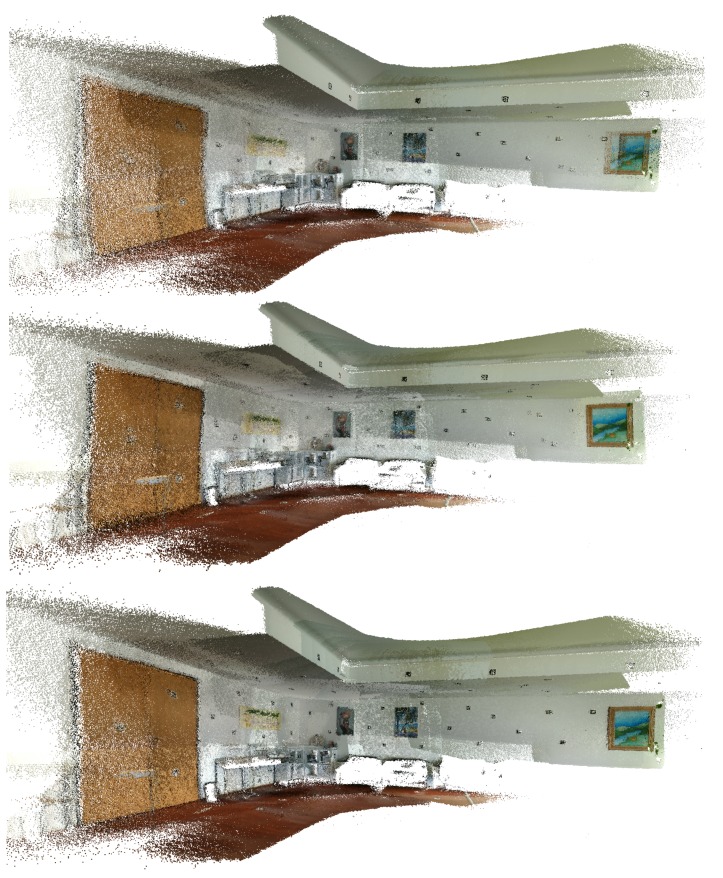
*MeetingRoom* dataset coloured with texture obtained from original images: (**top**) Google Tango; (**middle**) Iterative Closest Point (ICP); (**bottom**) Our approach.

**Figure 20 sensors-20-01497-f020:**
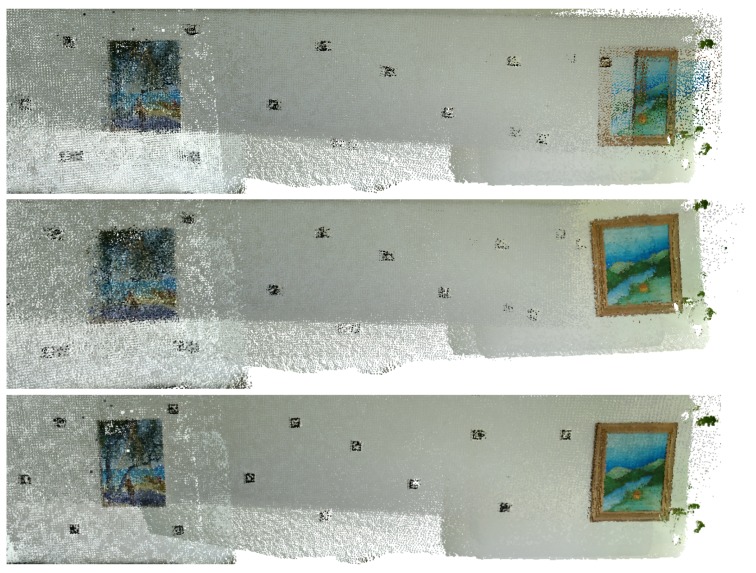
*MeetingRoom* dataset coloured with texture obtained from original images (detail 1): (**top**) Google Tango; (**middle**) ICP; (**bottom**) Our approach.

**Figure 21 sensors-20-01497-f021:**
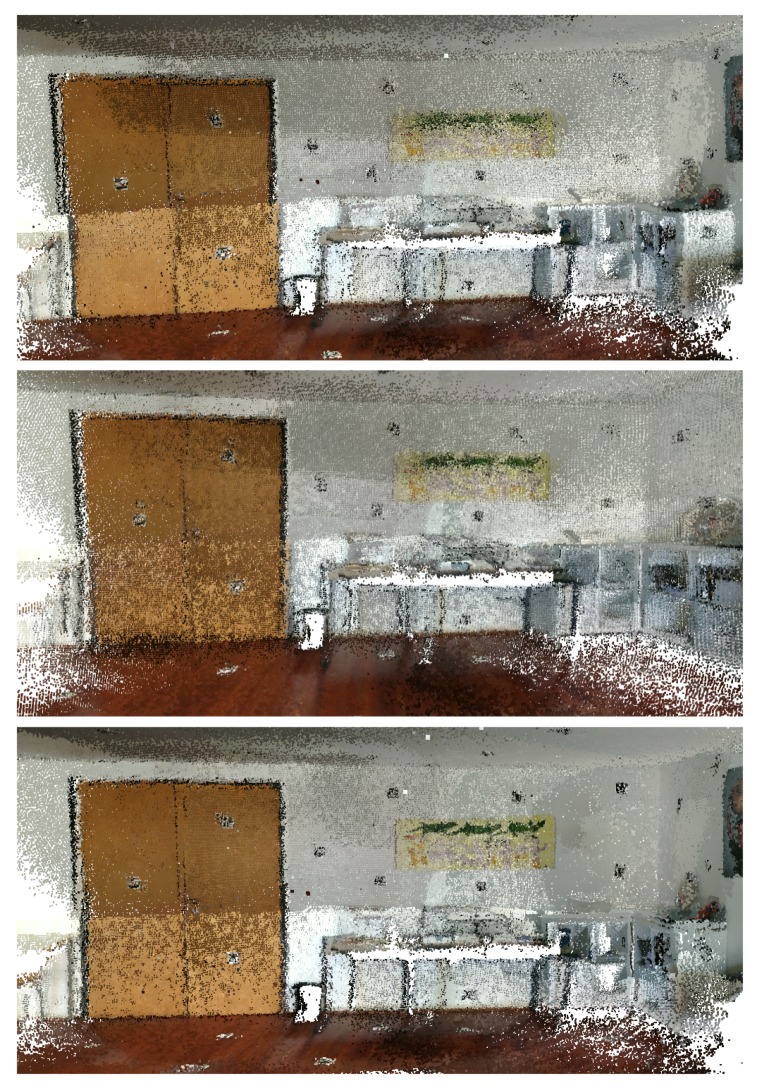
*MeetingRoom* dataset coloured with texture obtained from original images (detail 2): (**top**) Google Tango; (**middle**) ICP; (**bottom**) Our approach.

**Figure 22 sensors-20-01497-f022:**
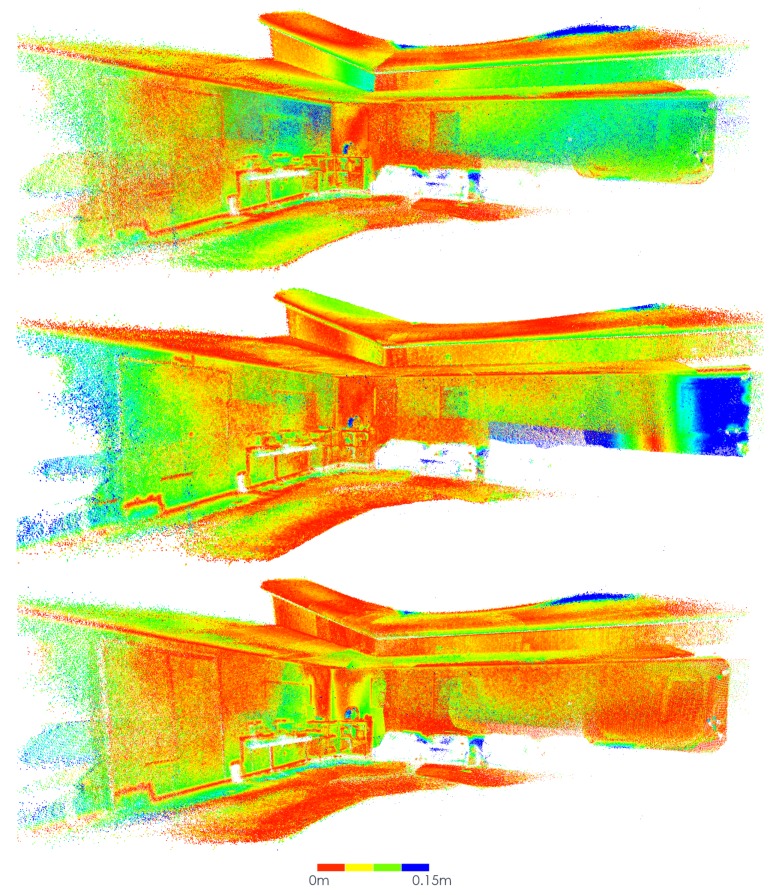
*MeetingRoom* dataset colourmapped with Hausdorff distance to FARO laser scanned point cloud: (**top**) Google Tango; (**middle**) ICP; (**bottom**) Our approach.

**Figure 23 sensors-20-01497-f023:**
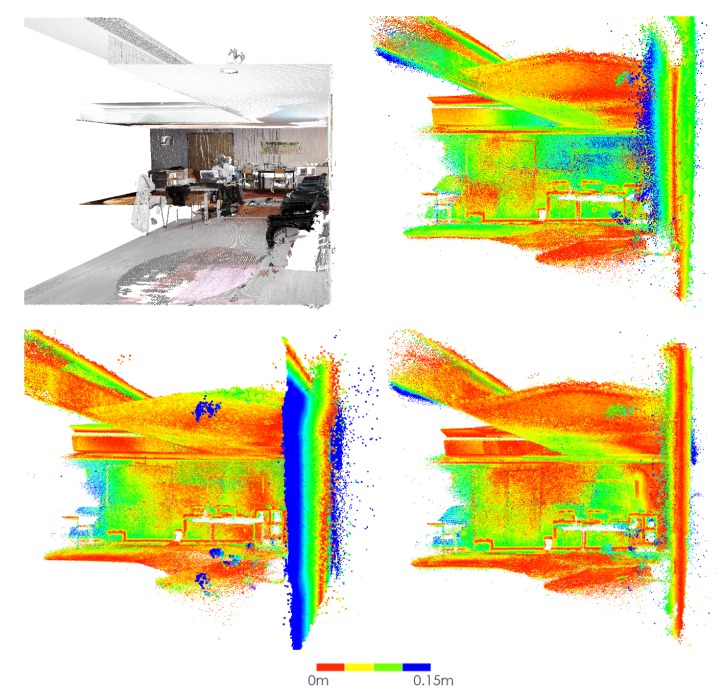
*MeetingRoom* dataset colourmapped with Hausdorff distance to FARO laser scanned point cloud (detail): (**top-left**) Groud truth laser scan; (**top-right**) Google Tango; (**bottom-left**) ICP; (**bottom-right**) Our approach.

**Figure 24 sensors-20-01497-f024:**
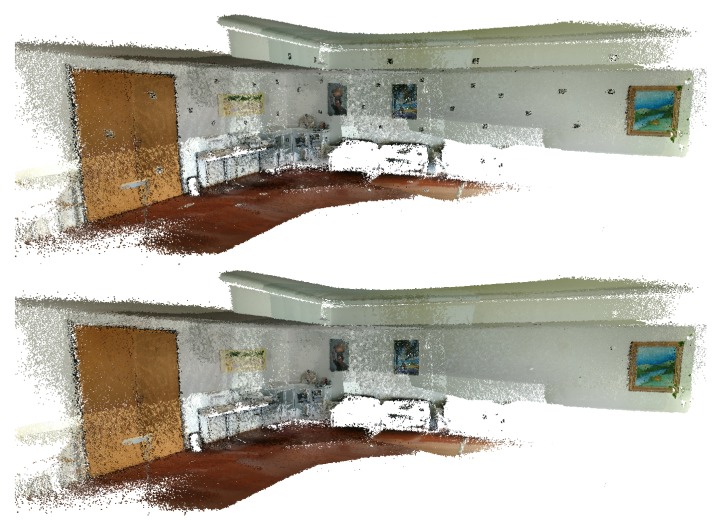
*MeetingRoom* dataset coloured with texture: (**top**) obtained from original images; (**bottom**) obtained from restored images.

**Figure 25 sensors-20-01497-f025:**
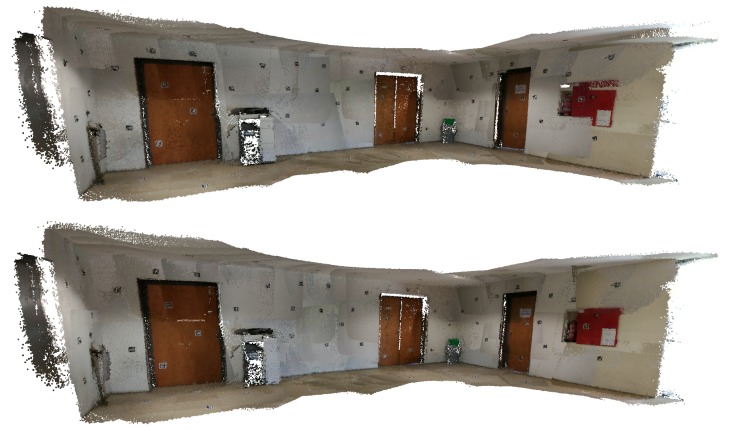
*Lobby* dataset coloured with texture obtained from original images: (**top**) Google Tango; (**bottom**) Our approach.

**Figure 26 sensors-20-01497-f026:**
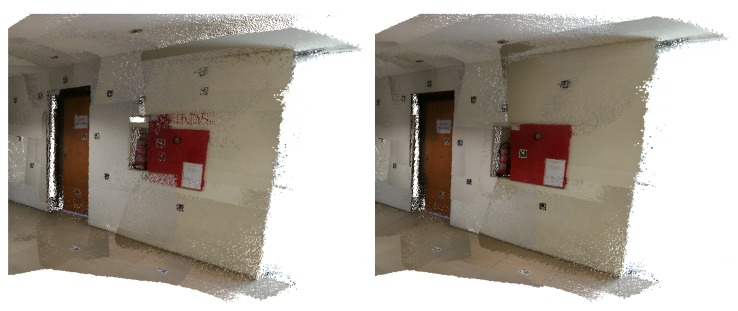
*Lobby* dataset coloured with texture obtained from original images (detail): (**left**) Google Tango; (**right**) Our approach.

**Figure 27 sensors-20-01497-f027:**
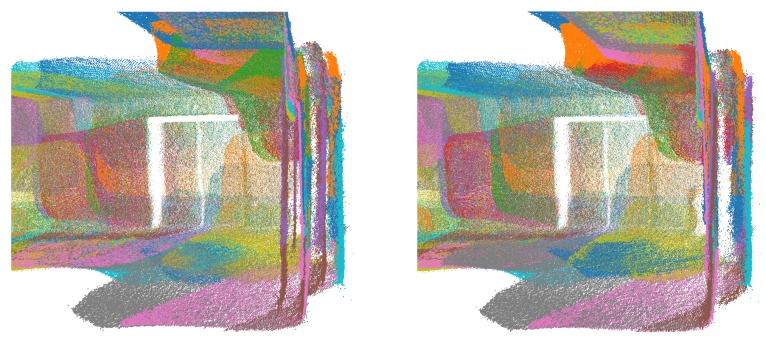
*Lobby* dataset point clouds coloured using a colourmap: (**left**) Google Tango; (**right**) Our approach.

**Figure 28 sensors-20-01497-f028:**
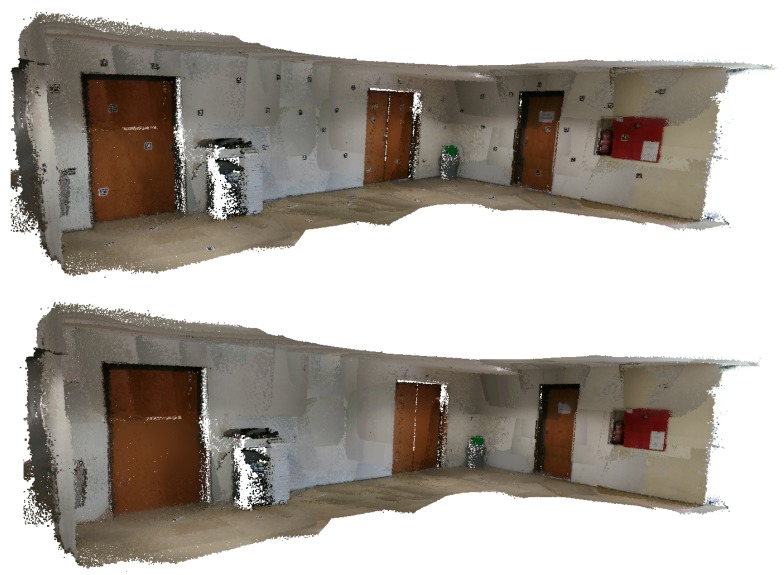
*Lobby* dataset coloured with texture: (**top**) obtained from original images; (**bottom**) obtained from restored images.

**Table 1 sensors-20-01497-t001:** 3D alignment processing times for the *MeetingRoom* dataset.

	Time (mm:ss)
**ICP**	03:02
**Our Approach**	03:50

**Table 2 sensors-20-01497-t002:** Hausdorff distance from the *MeetingRoom* dataset to the FARO laser scanned point cloud.

	Mean (m)	RMS (m)	min (m)	max (m)
**Google Tango**	0.0337	0.0442	0.0001	0.1500
**ICP**	0.0259	0.0372	0.0001	0.1500
**Our Approach**	0.0247	0.0324	0.0001	0.1500

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
