# Peer review of "Enhancement of RGB-D Image Alignment Using Fiducial Markers"

_sensors, 2020, doi:10.3390/s20051497_

Round 1

Reviewer 1 Report

The authors present an approach for RGB-D image alignment using fiducial markers. The procedure is capable of enhancing 3D models using consumer-level RGB-D cameras.

  • Include more details in Texture refinement and marker removal section. Talk about the blurring factor, how is this value set? Does it depend on the frame size? How is set the patch size around the markers? 
  • In the Experiment Section, describe the results. Is there any limitation? How about a comparison with other methods? Will the proposal work with other datasets?

Author Response

“Include more details in Texture refinement and marker removal section. Talk about the blurring factor, how is this value set? Does it depend on the frame size? How is set the patch size around the markers?"

We agree with the reviewer in that we should provide more details concerning the parameterization of the proposed texture refinement methodology. To address this, more detail has been included in Section 4 (Texture refinement and marker removal), along with the values for the blurring factor, this explanation: 

“The values for the blurring factor, along with the size of the marker objects and patch size around them, can be modified through command line arguments or assigned easily in the code. These values were empirically set because they depend on various distinct factors, the characteristics of the scene, and the sensors used.”

We initially did not provide more detail on the texture restoration technique since we focused more on the geometric enhancement. Texture restoration was used merely as a way to mitigate the downside of the proposed approach, which requires the scene to be polluted with markers.

“In the Experiment Section, describe the results. Is there any limitation? How about a comparison with other methods? Will the proposal work with other datasets?”

As per the reviewer’s request, section 5 (the Experiment Section) has been updated with information about the results, including some discussion about what were considered the main limitations. We would like to make note that the manuscript provides in sections 3 and 5 several qualitative and quantitative results in three different datasets (Example/Office, MeetingRoom, and Lobby). Thus, we are convinced that the approach is robust and should work for other datasets as well. Taking this into account we have added a sentence in the conclusions section as follows: 

“Results presented include several qualitative and quantitative assessments in three different datasets. Thus, we are convinced that the approach is robust and should in principle work for other datasets. “ 

Our results showcase the improvement of registration when compared with the Google Tango reconstructions, through visual and quantitative results (Hausdorff distance). Google Tango is widely used and well documented. It is based on VSLAM and motion tracking, and performs a bundle adjustment optimization as part of its algorithm. As such, we think this is a good way to demonstrate our results. Nevertheless, we recognize the room for comparison with other methods and, taking into consideration the reviewer’s request and reviewer 2’s suggestion of comparison with ICP, have added a comparison with ICP to the results section.

Reviewer 2 Report

This paper presents an indoor point cloud alignment algorithm based on fiducial markers. Moreover, the authors also present a post-processing method to remove these markers from the texture of the scene. Results show that the proposed point cloud alignment method can reduce alignment errors between captures. The markers used to enhance the alignment also can be removed seamlessly from the final model texture. The major problem of this paper is that the proposed method lacks of novelty. Actually, the proposed method is developed based on two existing tools, one is the aruco marker detection tool, and the other one is the image inpainting tool. Therefore, I think this paper is more like an engineering application article than an academic research paper.

Moreover, there are several limitations on the proposed method. The proposed method is more like a refinement process to improve an initial point cloud registration result obtained from a SLAM system, such as Google Tango platform. Therefore, the proposed method requires an initial guess of the camera poses and each aruco marker position. In fact, it is difficult to meet these requirements in many practical applications.

In Section 5, the authors should present the processing time of the proposed method. Moreover, the authors also need to compare the proposed method with other existing registration refinement algorithms, such as ICP, GICP, NICP, etc.

Finally, the following highly related reference is missing.

CY Tsai and CH Huang, “Indoor scene point cloud registration algorithm based on RGB-D camera calibration,” Sensors 2017, 17(8), 1874.

Author Response

“The major problem of this paper is that the proposed method lacks of novelty. Actually, the proposed method is developed based on two existing tools, one is the aruco marker detection tool, and the other one is the image inpainting tool. Therefore, I think this paper is more like an engineering application article than an academic research paper.”

We thank the reviewer for their analysis. It has led us to believe that we should have explained the focus of the manuscript, as well as its core contributions, more clearly. In the next few lines we try to clarify the matter:

The main novelty of the proposed method is to enhance the geometry of 3D reconstructions by improving the registration of the captures (not the marker detection nor the inpainting processes, which are necessary, but not the main focus of the work). One of the main limitations of the registration step is the robust matching of keypoints. The proposed methodology uses fiducial markers, as these are robustly detected and identified, to improve feature detection and allow for a better alignment of 3D point clouds. Several other fiducial markers might have been selected since the optimization procedure is agnostic to this. Aruco Markers were used due to the robust detection already available but the estimation of the position of the markers could have been performed another way, as suggested by the reviewer. 

We consider that the developed methodologies create significant additional value by combining existing techniques to create a relevant system, also from a scientific standpoint. Moreover, the proposed method, while integrating the mentioned tools, is not strictly dependent on them, or based off of them.

The texture restoration procedure uses the image inpainting tool as a step to achieve the demonstrated results. This is because it was seen as the obvious first step to restore an area. However, the developed pipeline adds considerable value, producing usable results, which the inpainting tool itself does not. Furthermore, we show the successful restoration of texture in areas where the marker detection capture would not work, such as the borders of an image. On the other hand, as stated before, the focus of the manuscript is on the geometric enhancement, and the texture restoration was used merely as a way to mitigate a downside of the approach taken.

We have rewritten several parts of the manuscript in order to make the focus and contributions more clear.

“...there are several limitations on the proposed method. The proposed method is more like a refinement process to improve an initial point cloud registration result obtained from a SLAM system, such as Google Tango platform. Therefore, the proposed method requires an initial guess of the camera poses and each aruco marker position. In fact, it is difficult to meet these requirements in many practical applications.”

We thank the reviewer for their comment. The main objective of this work was to develop an optimization procedure capable of enhancing 3D reconstructions created using a hand-held RGB-D camera. However, while the optimization procedure does require an initial guess of the camera poses, our methodology is able to produce one, if it is not available. The initial poses of the cameras were read from the Google Tango datasets because our objective was to refine them, however it is important to note that in the absence of a SLAM system producing the initial reconstruction, our system can still function. In light of the review’s observation, we have added the following information to the manuscript, in section 3:

“However, if this data is not available, our system is able to produce one by utilizing the detected markers. In this case, one of the cameras is considered as the World reference frame and the transformations from each camera to the World are calculated using chains of transformations between markers and cameras, taking advantage of common marker detections between cameras.“

“In Section 5, the authors should present the processing time of the proposed method.”

The processing times have been added in Section 5, along with some detail of the system used to perform the experiments.

“... the authors also need to compare the proposed method with other existing registration refinement algorithms, such as ICP, GICP, NICP, etc.”

Our results showcase the improvement of registration when compared with the Google Tango reconstructions, through visual and quantitative results (Hausdorff distance). Google Tango is widely used and well documented. It is based on VSLAM and motion tracking, and performs a bundle adjustment optimization as part of its algorithm. As such, we think this is a good way to demonstrate our results. Nevertheless, we recognize the room for comparison with other methods and, taking into consideration the reviewer’s suggestion of comparison with ICP, along with reviewer 1’s request, we have added that comparison to the results section.

“Finally, the following highly related reference is missing.

CY Tsai and CH Huang, “Indoor scene point cloud registration algorithm based on RGB-D camera calibration,” Sensors 2017, 17(8), 1874.”

We thank the reviewer for their suggestion. The paper is clearly relevant and on topic. A reference to the paper was added.

Reviewer 3 Report

In the submitted manuscript to review Enhancement of RGB-D Image Alignment Using Fiducial Markers’’ in face of the socio-economic and environmental aspects of its cultivation and marketization" the title reflect the content and emphasize the paper's interest and significance and the abstract explain the significance of the paper and properly address the background.

The ideas, methods and results presented in this paper are supported by literature contributions and a critical discussion is provided. All the figures and tables provided in manuscript are required, well explained and justified. Overall, English language is acceptable.

In general, this is an interesting manuscript. It is well conceived. The topic is current and it fits into trends in science as well as in the manufacturing practice. In my opinion, the article is suitable for publication in the scientific journal Sensors in present form.

Author Response

“In the submitted manuscript to review Enhancement of RGB-D Image Alignment Using Fiducial Markers’’ in face of the socio-economic and environmental aspects of its cultivation and marketization" the title reflect the content and emphasize the paper's interest and significance and the abstract explain the significance of the paper and properly address the background.

The ideas, methods and results presented in this paper are supported by literature contributions and a critical discussion is provided. All the figures and tables provided in manuscript are required, well explained and justified. Overall, English language is acceptable.

In general, this is an interesting manuscript. It is well conceived. The topic is current and it fits into trends in science as well as in the manufacturing practice. In my opinion, the article is suitable for publication in the scientific journal Sensors in present form.”

We would like to take this opportunity to thank the reviewer for their feedback.  Also, we make note of the reviewer's recognition of the relevance of this manuscript to the topic of 3D reconstruction.

Round 2

Reviewer 2 Report

The authors have improved the manuscript according to the review comments. I still have one comment. In the authors’ response, they say the processing times have been added in Section 5. However, the only relevant information is shown on Page 15, Line 354, and the authors only state that their approach is overhead of time. I don’t think this explanation is clear enough. Instead, it is better to provide a table similar to Table 1 on Page 19 to compare the processing time between these three methods. My point is that real-time performance is also one of the key factors in evaluating the advantages of point cloud alignment algorithms.

Author Response

“In the authors’ response, they say the processing times have been added in Section 5. However, the only relevant information is shown on Page 15, Line 354, and the authors only state that their approach is overhead of time. I don’t think this explanation is clear enough. Instead, it is better to provide a table similar to Table 1 on Page 19 to compare the processing time between these three methods. My point is that real-time performance is also one of the key factors in evaluating the advantages of point cloud alignment algorithms.”

We thank the reviewer once again for their comments. We have included additional information regarding processing times on Page 15, Line 354 (now 356). In this paragraph we point out some limitations of our approach, namely the fact that our approach requires time and logistics to prepare the environment before any capture. The time necessary for this preparation is very dependent on the user, the scene, and the density of markers desired. In the last section of the manuscript we argue that further studies are necessary to evaluate the ideal density of markers for our optimization in order to provide guidelines regarding marker density and placement to optimize preparation time and alignment results.

On Page 15, Line 341 we had added the number of captures of the dataset and the time it took for the optimization to execute, as per the reviewer’s previous request of presenting the processing time of the proposed method, however we understand the added value of directly showcasing it next to the processing time of other methods. As such, we have added a table on Page 17 presenting the processing time of our method compared with ICP, along with some considerations about Google Tango, which justify why the time for this approach cannot be accurately measured. Google Tango is a real time reconstruction method performing bundle adjustment optimization during the capture process, while our approach, being a post processing method, only executes after all the data has been captured. Even so, Google Tango does require some time, after data acquisition, to produce the final reconstructed model. Unfortunately, it is difficult to evaluate precisely tango processing time since the system is not open source and the time we could evaluate would also include other processes besides optimization, such as mesh creation and processing of storage data changes.

The following sentences were added to the manuscript:

[Google Tango] "... performs bundle adjustment optimization during the capture process and thus it is not directly comparable with our method that only executes the optimization after all the data has been captured. It is noteworthy that Google Tango also requires additional time after data capture to produce the final reconstructed model. Unfortunately, it is difficult to evaluate precisely tango processing time since the system is not open source and the time we could evaluate would also include other processes besides the optimization, such as mesh creation and processing of stored data changes.”